# Cloning, expression, and molecular modification of glycoside hydrolase family 5 genes from *Thermoascus aurantiacus*

Hongwei Shao *

School of Life Sciences, Qilu Normal University, Ji'nan, China

* hongwei_80@sina.com

## Abstract

In this paper, a novel bifunctional cellulase gene *cel1* was cloned from *Thermoascus aurantiacu*s by PCR and heterologously expressed in *Pichia pastoris* GS115. Bioinformatics and other related tools were used to compare the nucleotide homology of target genes, and analyze the signal peptide, transmembrane domain, hydrophilicity, secondary and tertiary structure of proteins. It was concluded that *cel1* has similar endoglucanase nucleotide sequences and falls under the GH5 family. It was also found that *cel1* has nucleotide sequences similar to glucosidase, which can infer that *cel1* may have the properties of glucosidase, indicating that *cel1* is multifunctional. At the same time, a part of the nucleotide sequence of the gene was removed to obtain a new gene *cel2*, and after highly efficient heterologous expression, its specific activity was found to be 2.1 times higher. Its enhancement is related to the exposure of the protein's hollow three-dimensional structure. This paper provides good material for exploring the relationship between the structure of bifunctional enzymes and their functions, which lays a solid foundation for further research and applications, and provides useful insight for gene mining of other novel enzymes.

**Data Availability Statement:** All relevant data are within the manuscript and its Supporting Information files.

**Funding:** The authors received no specific funding for this work.

## Introduction

Cellulose is a macromolecular polysaccharide composed of glucose, insoluble in water and common organic solvents. It is the main component of plant cell walls and the most widely distributed as well as the most abundant polysaccharide in nature, accounting for more than 50% of the carbon content of the plant kingdom [1]. The base ring of cellulose macromolecules is a macromolecular polysaccharide composed of D-glucose and beta-1,4 glycosidic bonds, and its chemical composition contains 44.44% of carbon, 6.17% of hydrogen, and 49.39% of oxygen [2]. Microorganisms that decompose cellulose include fungi and bacteria, and the cellulase system in fungi is thought to include three hydrolases, that is: (i) endoglucanase, which usually randomly cuts the amorphous part of celluloses; (ii) exoglucanase, which releases cellobioses from the non-reducing or reducing ends, usually from the crystalline part of celluloses; (iii) β-glucosidase, which releases glucose from cellobioses and short-chain cello-oligosaccharides [3]. All these enzymes have potential applications in various fields such as textile, paper, and detergent industries and protoplasts production [4, 5].

**Competing interests:** The authors have declared that no competing interests exist.

In recent years, thermophilic cellulolytic fungi have attracted considerable research interest, because they can produce thermostable cellulases. These enzymes can be used in many bio-technological applications. Cellulases produced by thermophilic fungi have high activity and stability under high-temperature conditions. Furthermore, similar to mesophilic fungi, cellulases produced by thermophilic fungi usually contain various components. *Thermoascus aurantiacu*s is a fungus widely distributed in the soil that produces thermostable enzymes. Analysis of the *T. aurantiacus* genome revealed that it can encode a number of cellulolytic enzymes, including an endoglucanase belonging to the glycoside hydrolase family 5 (GH5) and two beta-glucosidases belonging to the glycoside hydrolase 3 family (GH3).

Present study involves a thermophilic fungus *Thermoascus aurantiacus* [6], which has been known to produces several plant cell wall degrading enzymes including endoglucanases [7–9]. It is well known that *T. aurantiacus* is an organism which can produce all three cellulases [10–14]. The endoglucanase exhibits high rates of substrate hydrolysis, superior thermostability, remarkable stability over a wide range of pH values and has tremendous commercial potential [15–17]. The endoglucanases find applications in increasing the yield of fruit juices, beer filtration, oil extraction, improving the nutritive quality of bakery products and animal feed, in enhancing the brightness, smoothness, and overall quality of cellulosic garments and in producing fungal protoplast and hybrid strains [15, 18]. The quantity of research data described above shows importance of this endoglucanase from *T. aurantiacus*.The level of enzyme activity directly affects the production cost and production efficiency of industrial production. In most cases, the activity of the enzyme is too low to be suitable for commercial and therapeutic use, so increasing the catalytic activity of the enzyme is an important research direction for the directed evolution of the enzyme. There are many reports that have demonstrated that the directed evolution techniques have a good effect on increasing the catalytic activity of enzymes [19]. In this study, a gene encoding a bifunctional enzyme with cellulase activity was cloned from *T. aurantiacus*, a highly expressed *Pichia pastoris* genetically engineered strain was constructed, and its enzymatic properties were systematically studied. The gene was molecularly engineered to obtain a new enzyme with enhanced activity.

## Materials and methods

### Bacterial strains, plasmids, and growth conditions

*T. aurantiacus* IMI 216529 was obtained from the Centre for Agriculture and Biosciences International (CABI) located in Egham, Surrey (UK). *Escherichia coli* Trans1-T1 used for gene cloning was purchased from Sangon, Shanghai. The expression vector pPIC9K was purchased from Invitrogen (USA). *Pichia pastoris* GS115 was used as a host for cellulase expression. To induce cellulase gene expression, *T. aurantiacus* mycelium grown in potato dextrose agar (PDA) plates was transferred to an induction medium [20]. For the induction of the enzyme, actively growing fungal mycelium was transferred from a potato dextrose agar plate to medium containing the following (g/l): lactose, 20.0;agar, 15.0; yeast extract, 4.0; $K_2 HPO_4 .^3H_2O$,1.0; $MgSO_4 .7H_2O$, 0.5; dissolved in distilled and tap water (3:1).Mycelia were collected for total RNA isolation after the fungus grew at 50˚C for 2 days.

### cDNA and genomic DNA cloning

After two days of incubation on induction medium at 50˚C, total *T. aurantiacus* RNA was extracted using the UNIQ-10 column TRIzol total RNA extraction kit. After the total RNA was treated with DNase I, the first strand of cDNA was synthesized using the RevertAid First Strand cDNA Synthesis Kit. The cDNA of *T. aurantiacus* was obtained by reverse transcription polymerase chain reaction (RT-PCR) [21]. Fungal cDNA was obtained by RT-PCR.Specific

primers were designed.cDNA was used as template and the target gene was amplified by PCR. The amplified fragment of interest was purified by electrophoresis, ligated into a pMD18-T easy vector, and transformed into chemically competent cells Trans-T1 for sequencing verification. The target gene was sent to Beijing Boshang Bioengineering Co., LTD for sequencing.

### Nucleotide sequence accession number

The nucleotide sequence accession number of *cel1* DNA gene of *T. aurantiacus* in GenBank is JC988729.1.

### Analysis of nucleotide and amino acid sequence

After sequencing, the correct gene sequence was obtained, and the nucleotide sequence and predicted amino acid sequence were analyzed by bioinformatics, involving the secondary and tertiary structure of nucleotides and proteins, and the hydrophilicity of enzymes was also predicted. Then, signal peptides and transmembrane domains, as well as the evolution of the enzyme system were analyzed. The bioinformatics analysis tools and related websites used in this section are shown in Table 1.

### Site-directed mutagenesis

Preliminary analysis was performed by NCBI site and enzyme homology comparison using the DNAMAN software. Several conserved sequences were found. The 3D structural analysis of proteins was then used to identify potential beneficial amino acids mutations.

3DLigandSite is a tool for predicting the likely interaction residues between monomers and ligands. According to the polarity of conservation, amino acid and the position of the spatial structure, such as whether it was in the curled position, and the feasibility of site-directed mutagenesis, some amino acids were selected. For example, E247A, E267A, E303A, E351A, and E376A were selected to mutate the Glu (E) of the amino acid to Ala (A); N68D to mutate the Asn (N) of the 68th amino acid to the Asp (D); N112D to mutate the Asn (N) of the 112th amino acid to the Asp (D); Y150H to mutate the Tyr (Y) of the 150th amino acid to the His (H); Y323H to mutate the Tyr (Y) of the 323rd amino acid to the His (H); N375D to mutate the Asn (N) of the 375th amino acid to the Asp (D).

Missense3D predicts the structural changes introduced by an amino acid substitution and is applicable to analyse both PDB coordinates and homology-predicted structures. For example, E247A is chosen because of potential polarity change.

The rapid mutagenesis kit from Quanzhou Gold Company was selected to site-directed mutagenes. According to the requirements of the instructions of the site-directed mutagenesis kit and the selected mutation site of the specific amino acid, primers are designed and

**Table 1. The bioinformatics analysis tools and related websites.**

| Analysis content | The bioinformatics analysis tools | Related websites |
|---|---|---|
| Gene Sequence Analysis | DNAMAN V6.0 | |
| Sequence Alignment | BLAST | http://www.ncbi.nlm.nih.gov/BLAST |
| Signal Peptide | SignalP 4.0 | http://www.cbs.dtu.dk/services/SignalP-4.0/ |
| Protein Specific Loci | Position | http://prosite.expasy.org/ |
| Glycosylation | Net OGlyc 4.0 | http://www.cbs.dtu.dk/services/Net OGlyc/ |
| Functional domain | SMART | http://smart.embl-heidelberg.de/ |
| Three-Dimensionalstructure | phyre2 | http://www.sbg.bio.ic.ac.uk/~phyre2/ |
| Binding site analysis | 3DLigandSite | http://www.sbg.bio.ic.ac.uk/3dligandsite/ |
| Mutation analysis | Missense3D | http://www.sbg.bio.ic.ac.uk/~missense3d/ |

synthesized Amino acids at specific positions can be mutated to other amino acids of different polarity. The recombinant expression vector plasmid was extracted in large quantity and subjected to SalI single-stranded linearization. The digested products were recovered and transformed into GS115 competent cells by electroporation and coated on the MD and MM plates.

### Construction of *cel1* mutants

A part of the nucleotide sequence of the gene *cel1* was removed to obtain a new gene *cel2* containing the nucleotide sequence 301–1236 in the gene *cel1* and with a nucleotide length of 936 bp. The gene *cel2* sequence was constructed into an expression vector and transformed into yeast through electroporation.

### Construction of pPIC9K/*cel1* and pPIC9K/*cel2* expression plasmids

The amplified DNA fragment was digested with SnaBI and NotI and ligated into the pPIC9K vector, which also had sites digested with SnaBI and NotI. The recombinant plasmids were named pPIC9K/*cel1* and pPIC9K/*cel2* and were subsequently confirmed by PCR, restriction digestion, and sequencing.

### Transformation of *P. pastoris* GS115

The recombinant expression vector pPIC9K/*cel1* or pPIC9K/*cel2* plasmid was extracted in large quantity and subjected to SalI single-stranded linearization. The digested products were recovered and transformed into GS115 competent cells by electroporation and coated on the MD and MM plates. After 2 to 4 days of incubation at 28°C, the monoclonal on the plate was extracted from the YPD plates containing different concentrations of limited G418 sulfate (Amresco), and was incubated upside down for 5 days at 28°C. Yeast transformants were screened for those with normal growth, and positive clones were identified by PCR amplification.

### Expression and purification of cel1 from *T. aurantiacus*

The selected transformants with the maximum enzymatic activity were cultivated in shake-flasks in (describe de media) and induced at (indicate the OD600) and the inducer(methanol) with the respective concentration and induction. BMGY medium [22] was also used for cell enrichment and BMMY medium [22] to induce enzyme production. The obtained fermentation broth was centrifuged at 120,000 r/min for 10 minutes at 4°C to obtain crude enzyme solution. The extracted supernatant was precipitated with 90% saturated ammonium sulfate for 24 hours, centrifuged at 8,000 r/min for 20 minutes. Then, the precipitate was collected, and re-dissolved with an appropriate amount of buffer, dialyzed for 24 hours, and the supernatant was applied to a Purification by Ni-NTA affinity chromatography, activity detection and enzyme purity by SDS-PAGE electrophoresis [23].

The slab gel with 1% gelatin after electrophoresis was washed in 50 mmol Tris-HCl (pH 8.0), containing 5% (v/v) Triton X-100, twice for 15 min at 4°C, followed by washing in the same buffer without Triton X-100 for 15 min to remove SDS. The gel was then incubated in 50 mmol Tris-HCl (pH8.0) for 12 h at 50°C to allow the degradation of the gelatin, and stained and destained in the same solution with SDS-PAGE.

### Activity assay and protein characterization

Cellulase activity: The activity of the enzyme was determined by the GHOSE method [12]. 100 μL of enzyme solution, 200 μL of 0.5% carboxymethylcellulose sodium (CMC-Na), and 100 μL of acetate buffer (0.4 mol/L, pH 5.0) were reacted at 50°C for 30 minutes, and 400 μL of

DNS reagent were added and heated for 10 minutes. The absorption was measured at a wavelength of 540 nm.The amount of enzyme required to produce 1 μmol of glucose per minute is the unit of enzyme activity (U).

Beta-glucosidase activity: 100 μL of enzyme solution, 200 μL of 1% salicin, and 100 μL of acetate buffer (0.4 mol/L, pH 5.0) were reacted at 50˚C for 30 minutes, and 400 μL of DNS reagent were added and heated for 10 minutes. The absorption measured at a wavelength of 540 nm and the amount of enzyme required to produce 1 μmol of glucose per minute is the unit of enzyme activity (U) [24].

## Glycoprotein detection

A protein glycosylation kit (Pierce R Glycoprotein Staining Kit, Thermo Scientific Company) was purchased to stain the target protein according to the instructions and observe whether it produces magenta bands.

The Pierce Glycoprotein Staining Kit is a convenient, fast, and sensitive colorimetric kit for staining glycosylated proteins in polyacrylamide gels or nitrocellulose membranes using the periodic acid-Schiff (PAS) method. The kit provides three essential reagents and a complete protocol to specifically stain glycosylated proteins (glycoproteins) that have been separated by polyacrylamide gel electrophoresis (PAGE). A gel or membrane containing separated proteins is treated with a periodate solution, which oxidizes cis-diol sugar groups in glycoproteins. The resulting aldehyde groups are detected through the formation of Schiff-base bonds with a reagent that produces magenta bands. Traditionally called the periodic acid-Schiff (PAS) reagent method, this staining technique is made easier by the convenient packaging of essential reagents in one kit with easy-to-follow instructions.

## Determination of the Michaelis constant Km of enzymes

Determination of the Michaelis constant Km of enzymes with temperature and pH value, and is independent of enzyme concentration, substrate concentration, or other relevant factors. It is an important means to identify enzymes under specific conditions. In this paper, the relationships between Km and the substrate concentration [S], the initial V of the enzyme, and the maximum reaction rate Vmax of the enzyme were summarized, that is, the Michaelis constant Km equation. According to this equation, the Michaelis constant of the corresponding enzyme was obtained.

## Optimal temperature for reactions

0.2 ml of substrates, 0.1 ml of acetate buffer with a concentration of 0.4 mol/L at pH 5.0, and 0.1 ml of enzyme solution were added to the new centrifuge tubes, respectively, and they were kept in 14 water baths with different constant temperatures at 20˚C, 25˚C, 30˚C, 35˚C, 40˚C, 45˚C, 50˚C, 55˚C, 60˚C, 65˚C, 70˚C, 75˚C, 80˚C, and 90˚C for 30 minutes. After that, 0.4 ml of DNS was added. After mixing, the tubes were placed in boiling water for 10 minutes before they were rinsed with cold water to room temperature. The absorbance of each reaction solution was measured at λ = 540 nm using a spectrophotometer, and the corresponding values were recorded. Experiments were repeated three times for each group to obtain an average value.

## Optimal pH value for reactions

0.2 ml of substrates, 0.1 ml of enzyme solution, and 0.1 ml of buffer with a concentration of 0.4 mol/L and a pH value of 2.0, 3.0, 4.0, 4.4, 5.0, 6.0, 7.0, 8.0, 9.0, and 10.0 were added to the new

centrifuge tubes, respectively. Then, they were kept in a constant temperature water bath at 50˚C for 30 minutes. After that, 0.4 ml of DNS was added. After mixing, the tubes were placed in boiling water for 10 minutes before they were rinsed with cold water to room temperature. The absorbance of each reaction solution was measured at $\lambda = 540$ nm using a spectrophotometer, and the corresponding values were recorded. Experiments were repeated three times for each group to obtain an average value.

### Effects of different metal ions on enzyme activity

After treating the enzyme solution with 2 concentrations (1 mmol/L and 10 mmol/L) of metal ions ($Ca^{2+}$, $Cu^{2+}$, $Hg^{2+}$, $Fe^{2+}$, $C0^{2+}$, $Mg^{2+}$, $Mn^{2+}$, $Zn^{2+}$, and $Ag^+$) and EDTA for 4 hours, 0.2 ml of substrates and buffer with a pH value of 5.0 were added into new centrifuge tubes in the enzyme reaction system. They were then placed in a constant temperature water bath at 50˚C for 30 minutes. The absorbance of each reaction solution was measured using a spectrophotometer(indicate the OD540), and the enzymatic activity was calculated and compared to that of the untreated enzyme solution, with the original enzyme activity at 100%.

### Effects of different organic solvents on enzyme activity

After treating the enzyme solution with 3 concentrations (1%, 15%, and 30%) of organic solvents (methanol, ethanol, isopropanol, and DMSO) for 4 hours, 0.2 ml of substrates and buffer with a pH value of 5.0 were added into the new centrifuge tubes. They were then placed in a constant temperature water bath at 50˚C for 30 minutes. The absorbance of each reaction solution was measured using a spectrophotometer, and the enzymatic activity was calculated and compared to that of the untreated enzyme solution, with the original enzyme activity at 100%.

### Study on salt tolerance

After treating the enzyme solution with 6 concentrations (0 mol/L, 0.5 mol/L, 1.0 mol/L, 1.5 mol/L, 2.0 mol/L, and 2.5 mol/L) of organic solvents for 4 hours, 0.2 ml of substrates and buffer with a pH value of 5.0 were added into the new centrifuge tubes. They were then placed in a constant temperature water bath at 50˚C for 30 minutes. The absorbance of each reaction solution was measured using a spectrophotometer, and the enzymatic activity was calculated and compared to that of the untreated enzyme solution, with the original enzyme activity at 100%.

### Data analysis

The effect of enzyme activity was analyzed by variance analysis, and the activity value of each enzyme under optimum conditions was analyzed by multiple comparison. All data were processed using statistical software SPSS 18.0

## Results

### Cloning analysis of *T. aurantiacus cel1* and *cel2*

After sequencing, the nucleotide length of the target gene *cel1* is 1365 bp (Fig 1). The ORF sequence of the gene contains 453 amino acids, and the codon usage rate is A: 265 (19.4%), C: 381 (27.9%), G: 452 (33.1%), and T: 267 (19.6%). The G + C content is 52.7%, and the molecular weight (kDa) are ssDNA: 422.93 and dsDNA: 841.67.

The nucleotide sequence of the target gene was analyzed using NCBI blast to predict its structure and family information, and the results showed that the cloned target gene *cel1* contained the nucleotide sequence of cellulase, which was named as cellulase gene *cel1*. It also

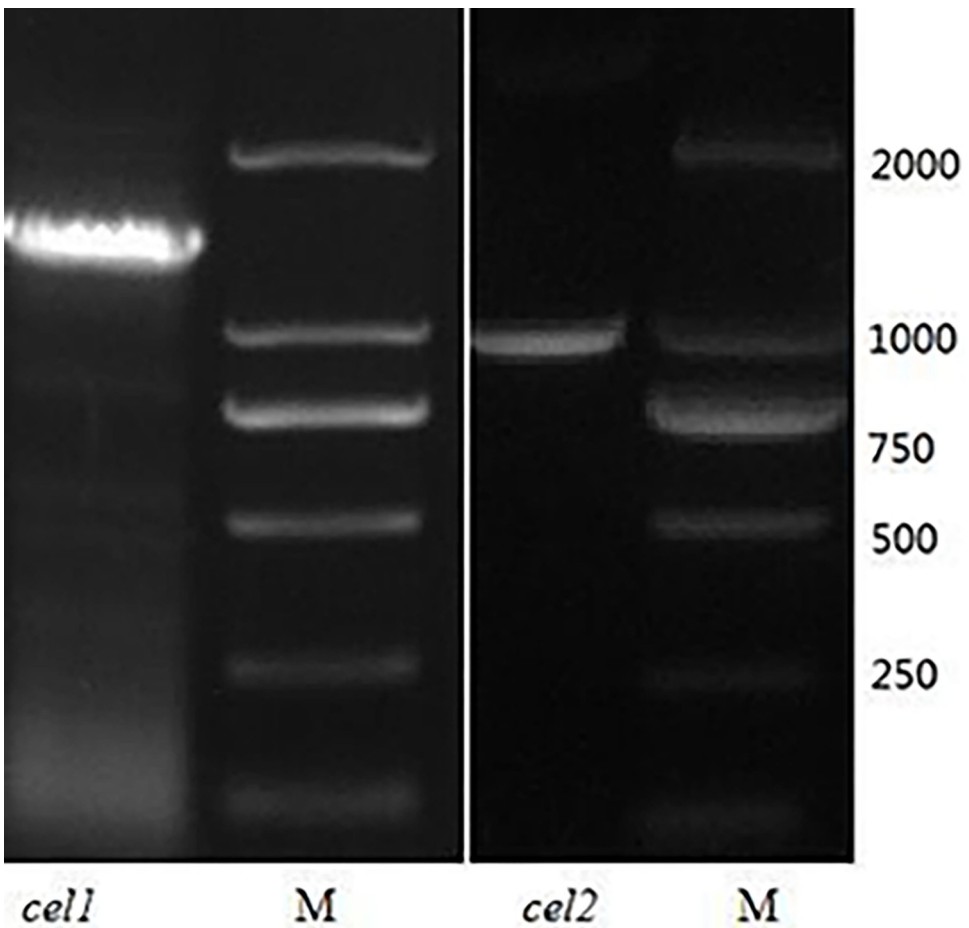

**Fig 1. Agarose gel electrophoresis of *cel1* and *cel2* from *T. aurantiacus* cDNA.** M. Trans2K DNA Marker;*cel1* and *cel2*: PCR amplification products with *T. aurantiacus* cDNA.

showed that *cel1* is 77% similar to endo-beta-1,4-glucanase [*Rasamsonia emersonii* CBS 393.64], 66% similar to glycoside hydrolase family 5 protein [*Oidiodendron maius* Zn], 60% similar to glycoside hydrolase family 5 protein [*Baudoinia panamericana* UAMH 10762], 62% similar to putative endoglucanase e1 protein [*Phaeoacremonium minimum* UCRPA7], and 61% similar to glycoside hydrolase family 5 protein [*Phialocephala scopiformis*] (Fig 2), which is presumed to own endocellulase properties and fall under the GH5 family. After further analyzing *cel1*, we found that the nucleotide sequence of 412–1236 (Fig 2) in the gene *cel1* is similar to cellulases of the GH5 family, and that there is a multi-domain region. The result showed that the nucleotide sequence of 301–915 (Fig 2) is similar to the nucleotide sequence of glucosidase, suggesting that *cel1* may have the properties of glucosidase and is multi-functional, which can be verified in the future.

The physicochemical properties of the amino acid of cellulase CEL1 were analyzed using the ExPASy ProParem Tool, and the results showed that the protein consisted of 453 amino acids with a molecular weight of 50,273.69. The O-linked glycosylation site of cellulase CEL1 was then analyzed by Net OGlyc 4.0 Server. It was found that CEL1 has O-linked glycosylation sites S42, S53, T54, T57, T62, T63, T88, and T93. Whereas, using protein-specific site analysis, no N-linked glycosylation sites were found.

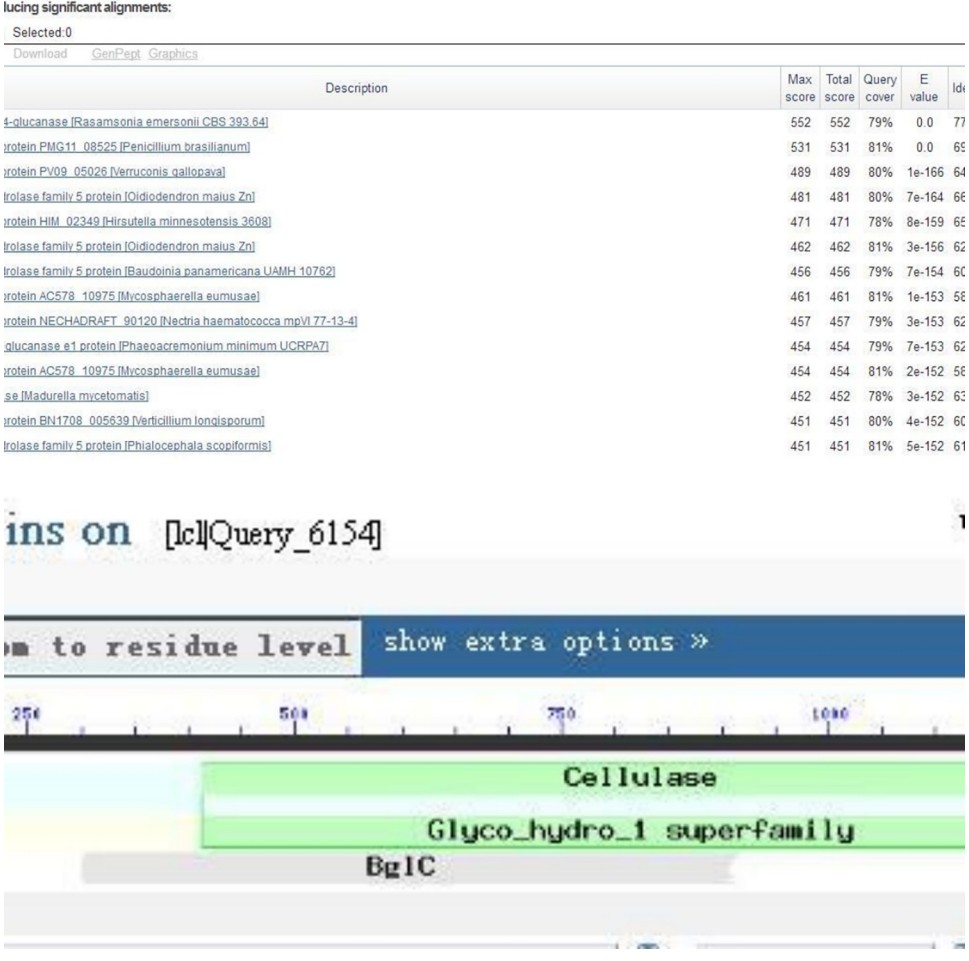

| Description | Max score | Total score | Query cover | E value | Id |
|---|---|---|---|---|---|
| 4-glucanase [Rasamsonia emersonii CBS 393.64] | 552 | 552 | 79% | 0.0 | 77 |
| protein PMG11_08525 [Penicillium brasilianum] | 531 | 531 | 81% | 0.0 | 69 |
| protein PV09_05026 [Verruconis gallopava] | 489 | 489 | 80% | 1e-166 | 64 |
| rolase family 5 protein [Oidiodendron maius Zn] | 481 | 481 | 80% | 7e-164 | 66 |
| protein HIM_02349 [Hirsutella minnesotensis 3608] | 471 | 471 | 78% | 8e-159 | 65 |
| rolase family 5 protein [Oidiodendron maius Zn] | 462 | 462 | 81% | 3e-156 | 62 |
| rolase family 5 protein [Baudoinia panamericana UAMH 10762] | 456 | 456 | 79% | 7e-154 | 60 |
| protein AC578_10975 [Mycosphaerella eumusae] | 461 | 461 | 81% | 1e-153 | 58 |
| protein NECHADRAFT_90120 [Nectria haematococca mpVI 77-13-4] | 457 | 457 | 79% | 3e-153 | 62 |
| glucanase e1 protein [Phaeoacremonium minimum UCRPA7] | 454 | 454 | 79% | 7e-153 | 62 |
| protein AC578_10975 [Mycosphaerella eumusae] | 454 | 454 | 81% | 2e-152 | 58 |
| se [Madurella mycetomatis] | 452 | 452 | 78% | 3e-152 | 63 |
| protein BN1708_005639 [Verticillium longisporum] | 451 | 451 | 80% | 4e-152 | 60 |
| rolase family 5 protein [Phialocephala scopiformis] | 451 | 451 | 81% | 5e-152 | 61 |

**Fig 2. Blast analysis of *cel1*.**

The functional domain of enzymes was analyzed using SMART. CEL1 was shown to have a signal peptide, a low complexity sequence, and a Pfam: Cellulase structure.

A part of the nucleotide sequence of the gene was removed to obtain a new gene *cel2* containing the nucleotide sequence 301–1236 in the gene *cel1* and with a nucleotide length of 936 bp (Fig 1). The physicochemical properties of the amino acid of CEL2 were analyzed using the ExPASy ProParem Tool, and the results showed that the protein consisted of 312 amino acids with a molecular weight of 34913.12.

## Expression and purification of recombinant protein

After transformation by electroporation, 96 recombinants of *P. pastoris* GS115 with pPIC9K/*cel1* and pPIC9K/*cel2* were coated on MD and MM plates. The cDNA was introduced into the yeast expression system, and positive clones were identified by PCR amplification. The supernatant was applied to a Purification by Ni-NTA affinity chromatography (Fig 3A, 3B).The target protein can be obtained by collecting the crest part in Fig 3A, 3B. Fig 3C shows SDS-PAGE analysis of the induced recombinant protein. Electrophoresis of CEL1 on SDS-PAGE showed a single band with a molecular weight of 70 kDa. However, it was slightly larger than that of the value calculated from the amino acid sequence. The molecular weight of CEL1 was

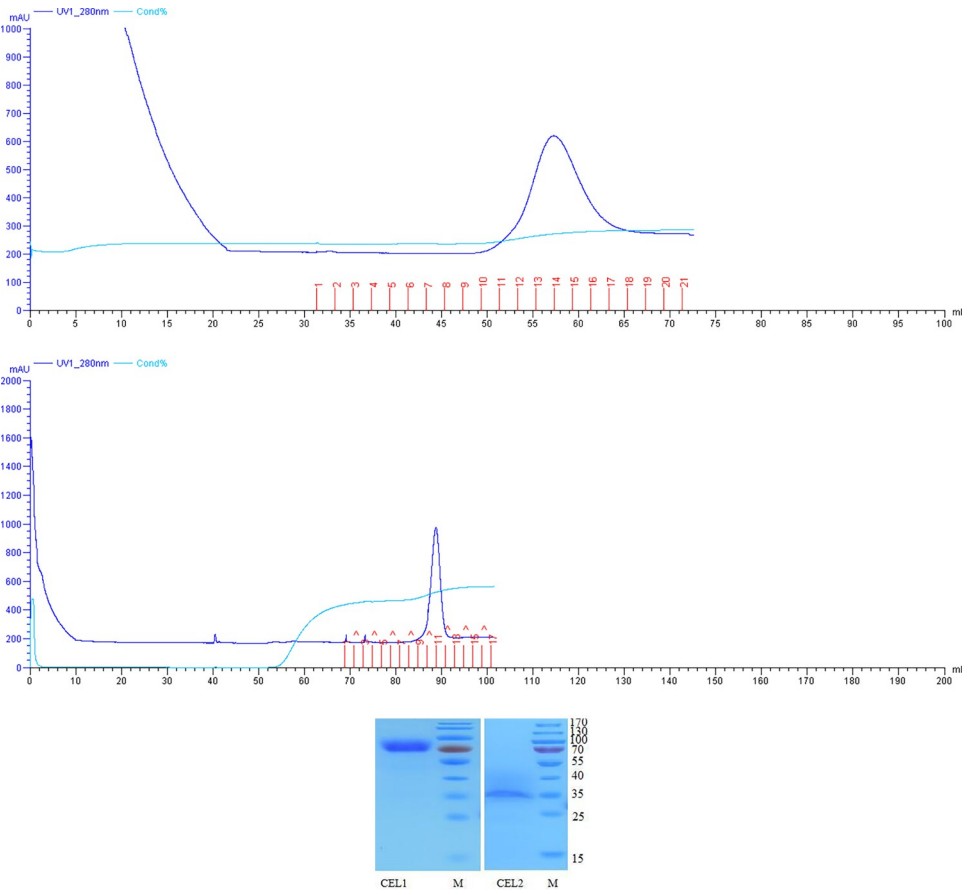

**Fig 3.** a. Nickel column affinity chromatography of CEL1. b. Nickel column affinity chromatography of CEL2. c. SDS-PAGE analysis of purified protein. M. marker(10-170kDa); CEL1 and CEL2:Purified products.

expressed in *P. pastoris*, which may be a result of O-linked glycosylation. A protein glycosylation kit was purchased to stain the target protein CEL1 according to the instructions and observe whether it produces magenta bands(Fig 4). One possible explanation is that glycosylation is associated with the formation of rigid structures that can increase the thermostability of many enzymes [25–27].

CEL2 was a purified expression product, which was a specific protein band of about 35 kDa. The analysis of physical and chemical properties showed that the theoretical molecular weight of CEL2 protein is relatively close to 34913.12, indicating that the protein CEL2 can be obtained by purification.

The purified mutant enzyme solutions were tested using SDS-PAGE purity test, and the results are shown in Fig 5. The enzyme solution showed only one protein band, indicating that mutant proteins can be obtained by purification.

## Michaelis constant Km of enzymes CEL1 and CEL2

According to the measurement curve of Michaelis constants of CEL1, the calculation formula for the Michaelis constant is y = 141.02x + 2.1264($R^2$ = 0.9972), which was calculated to be 66.67 mol/L. Whereas from the measurement curve of Michaelis constants of CEL2, the calculation formula for the Michaelis constant is y = 24.811x + 9.1104($R^2$ = 0.9901), which was calculated to be 2.73 mol/L.

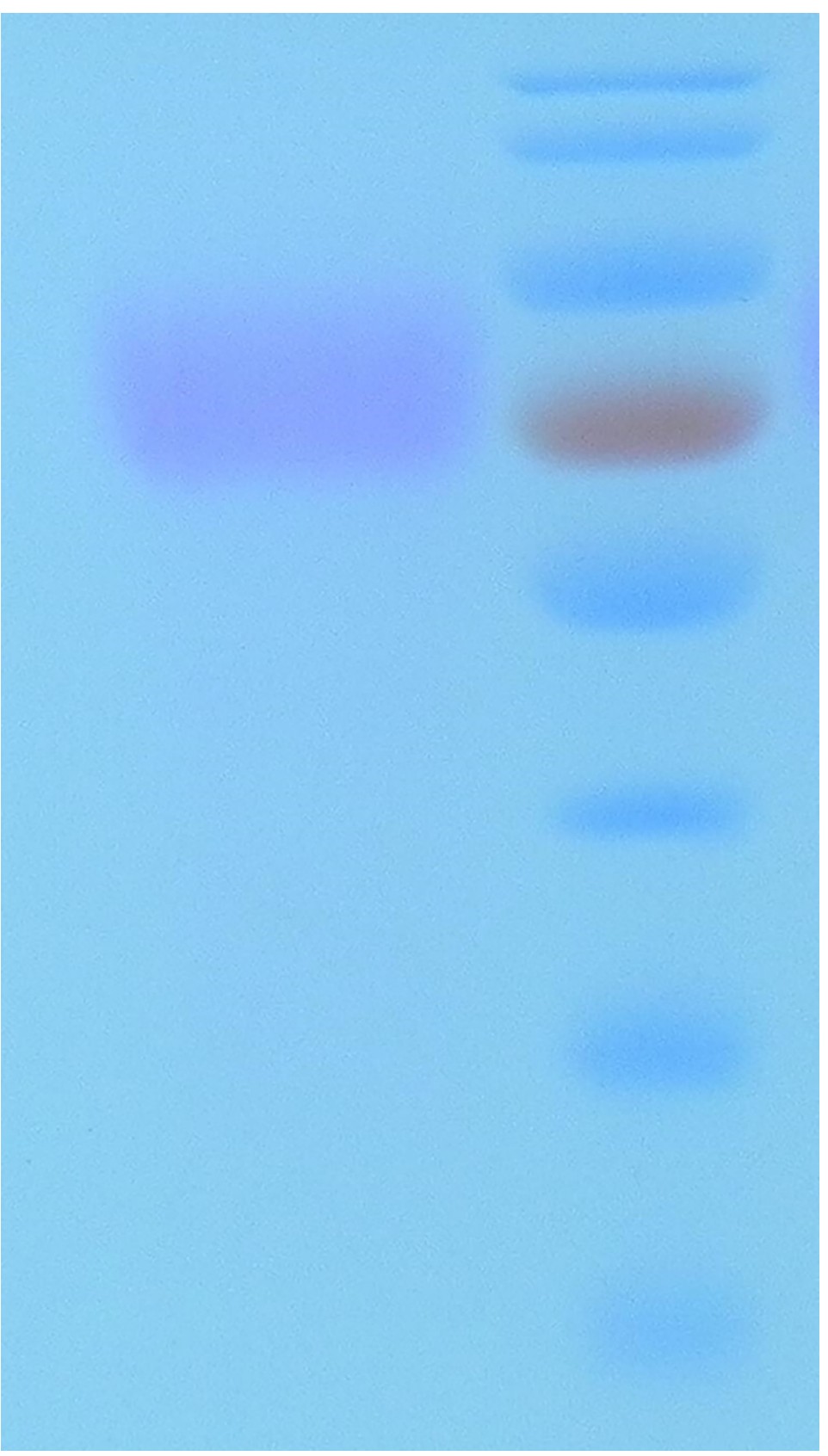

**Fig 4. SDS-PAGE analysis of stain protein.** M. marker(10-170kDa); CEL1: stain the target protein.

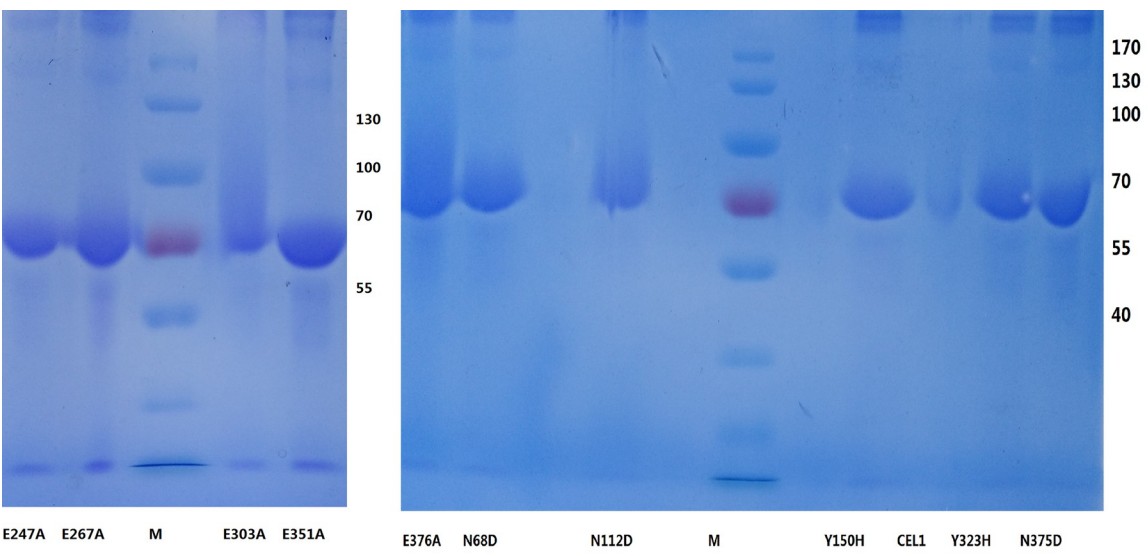

**Fig 5. SDS-PAGE analysis of purified mutation proteins.** M. marker(10-170kDa); E247A, E267A,E303A,E351A,N68D,N112D,Y150H, CEL1,Y323H,N375D:Purified products.

## Characteristics of CEL1 and CEL2

As a candidate GH5 endoglucanase, CEL1 exhibited a relative carboxymethyl cellulase specific activity of approximately 1.51 U/mg, and the enzyme showed degradation activity against salicin with specific activities of 0.52 U/mg. Meanwhile, the enzyme CEL2 showed degradation activity against sodium carboxymethyl cellulose and salicin with specific activities of 3.27 U/mg and 1.68 U/mg, respectively.

The enzymatic activities of CEL1 and CEL2 were measured at different temperatures, with the maximum enzymatic activity at 100%. The relative enzymatic activities of cellulase at different temperatures were also calculated, and the results are shown in Fig 6 below. The general trend of the temperature curve is to first increase then decrease. It can be seen that CEL1 has the strongest ability to decompose carboxymethylcellulose sodium (CMC-Na) and salicin at 55°C. About 85% of the maximum activity can be maintained at 60°C, while only 40% can be maintained at 65°C, indicating that CEL1 is not resistant to high temperatures, and its activity decreased significantly with increasing temperature. As can be seen from Fig 6 that CEL2 has the strongest ability to decompose carboxymethylcellulose sodium (CMC-Na) and salicin at 55°C, but at 60°C, only about 60% of the activity can be maintained, greatly reduced due to the increase of temperature, the activity was greatly reduced. At temperatures higher than 75°C, the differences are especially prominent, indicating that CEL2 is not resistant to high temperatures.

Cellulases react in buffer with different pH values under different pH conditions. The activities were measured under standard conditions, and relative enzymatic activities at different pH conditions were calculated with the maximum enzyme activity at 100%. The results are shown in Fig 7. The general trend of the pH value curve is to first increase then decrease. It can be seen that CEL1 has the strongest carboxymethylcellulose sodium (CMC-Na) and glucosidase activity at pH 5.0. Therefore, it can be inferred that the optimal pH value for CEL2 to decompose carboxymethylcellulose sodium (CMC-Na) is 5.0, indicating that CEL1 and CEL2 can easily decompose substrates in weak acid.

As can be seen in Fig 8, $Ag^+$, $Mn^{2+}$, and $Fe^{2+}$ can boost the carboxymethylcellulose sodium (CMC-Na) activity of CEL1. 1 mmol/L of $Ag^+$ had a significant effect, while 10 mmol/L of $Ca^2$

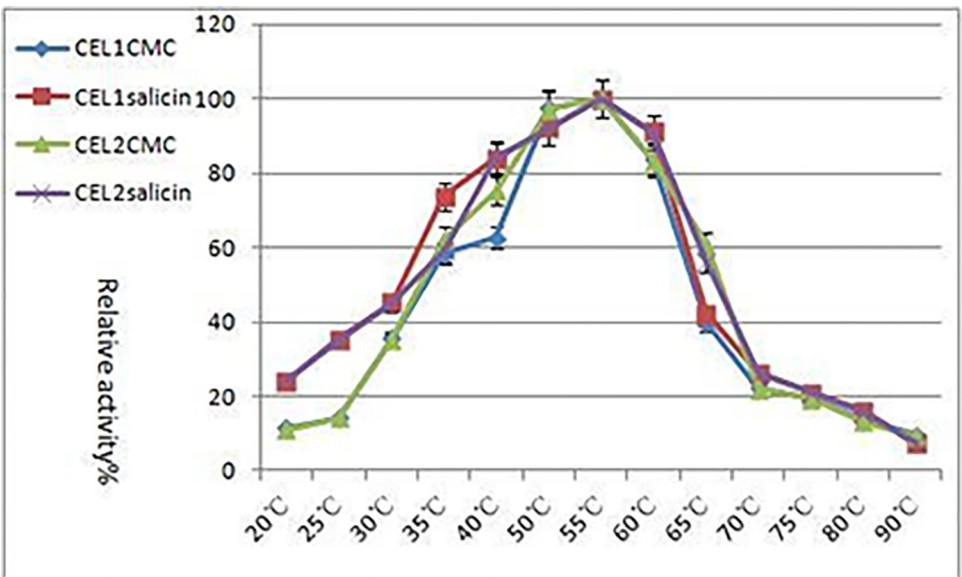

**Fig 6. The optimum reaction temperature.**

$^+$ had a relatively significant boosting effect. The inhibitory effects of $Mg^{2+}$, $Hg^{2+}$, and EDTA were obvious at the concentrations of 1 mmol/L and 10 mmol/L. The activity of 10 mmol/L EDTA was only 29.23% of the original enzymatic activity. From the effect of EDTA, it can be known that CEL1 depends on the activation effect of metal ions. Fig 8 also shows that $Ag^+$, $Mn^{2+}$, and $Fe^{2+}$ can boost the carboxymethylcellulose sodium (CMC-Na) activity of CEL2. 10 mmol/L of $Mn^{2+}$ had a significant effect, while 10 mmol/L of $Ca^{2+}$ showed significant boosting effects. $Mg^{2+}$, $Hg^{2+}$, $Cu^{2+}$, and EDTA had obvious inhibitory effects at the concentrations of 1 mmol/ L and 10 mmol/ L. The activity of 10 mmol/L EDTA was only 8.86% of the original enzymatic activity. After EDTA is bonded to metal ions, the activity is significantly reduced, indicating that CEL2 was more dependent on the activation of metal ions.

Methanol, ethanol, isopropanol, and DMSO at concentrations of 1% and 15% all showed relatively significant activation effects on the carboxymethylcellulose sodium (CMC-Na) CEL1 enzymatic activity, as shown in Fig 9. The activity is greatly improved especially when the concentration is at 1%. The enzymatic activities of isopropanol and DMSO increased the most, reaching 156.67% and 166% of the original enzymatic activity, respectively. At a concentration of 30%, all organic solvents showed significant inhibitory effects on CEL1 enzymatic activity. The activity of methanol decreased the most, which was only 63.33% of the original enzymatic

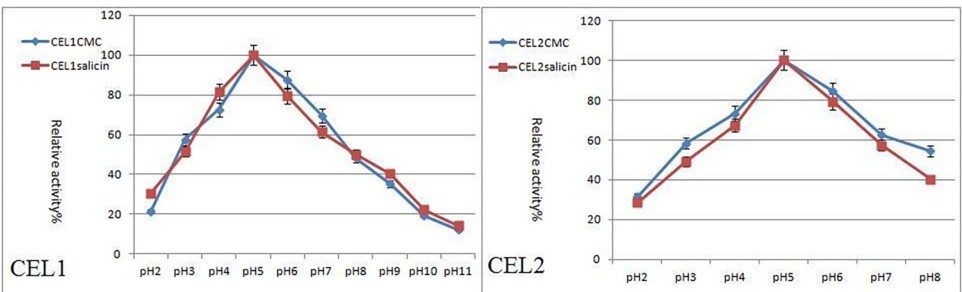

**Fig 7. The optimum reaction pH value.**

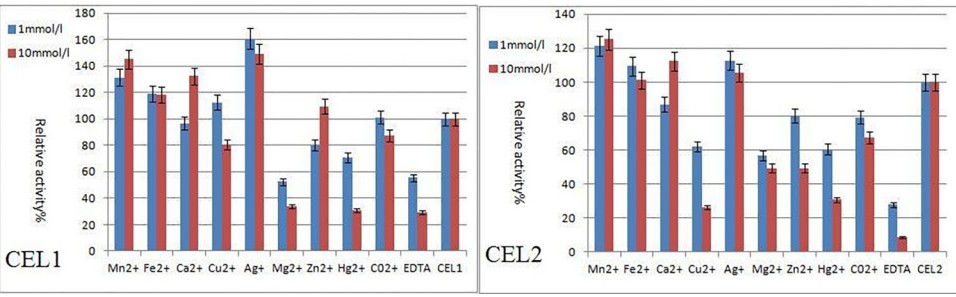

**Fig 8. Effect of metallic ions.**

activity. Fig 9 shows that the law of change of glucosidase activity was similar to that of cellulase activity. Organic substances may act on the enzymes with little effect on the substrates. Methanol, ethanol, isopropanol, and DMSO at concentrations of 1% and 15% showed inhibitory effects on CEL2 enzymatic activity. At 30% concentration, all organic solvents significantly inhibited CEL2 enzymatic activity, with isopropanol decreasing the most, which was less than 30% of the original enzymatic activity, as shown in Fig 9.

Fig 10 shows the enzymatic activities of CEL1 and CEL2 gradually decreasing with the increased NaCl concentration. When the concentration of NaCl was 2.5 mol/L, the carboxymethylcellulose sodium (CMC-Na) and salicin enzymatic activities of CEL1 remained above 60%, indicating that CEL1 has high salt tolerance. With the gradual increase of NaCl concentration, the enzymatic activity of CEL2 gradually decreased. When the NaCl concentration was 2.5mol/L, the CEL2 enzymatic activity remained above 51%.

## Comparison of mutant and CEL1 activity

The purified proteins were diluted to 0.1 mg/ml to measure the activity, and the results are shown in Fig 11. It was found that the mutation E303A and E376A substantially lost the

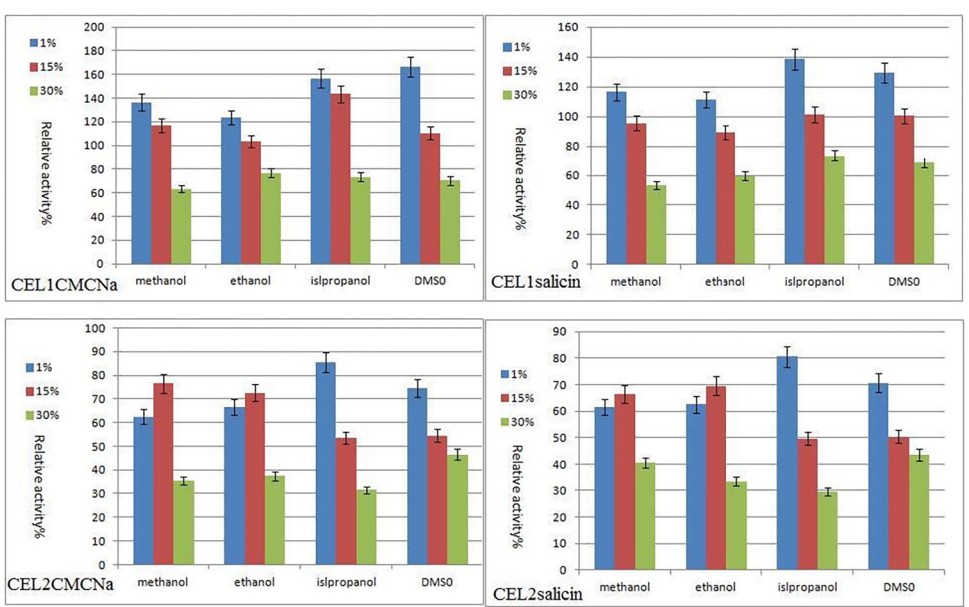

**Fig 9. Effect of organic solvents on enzymatic properties of CEL1 and CEL2.**

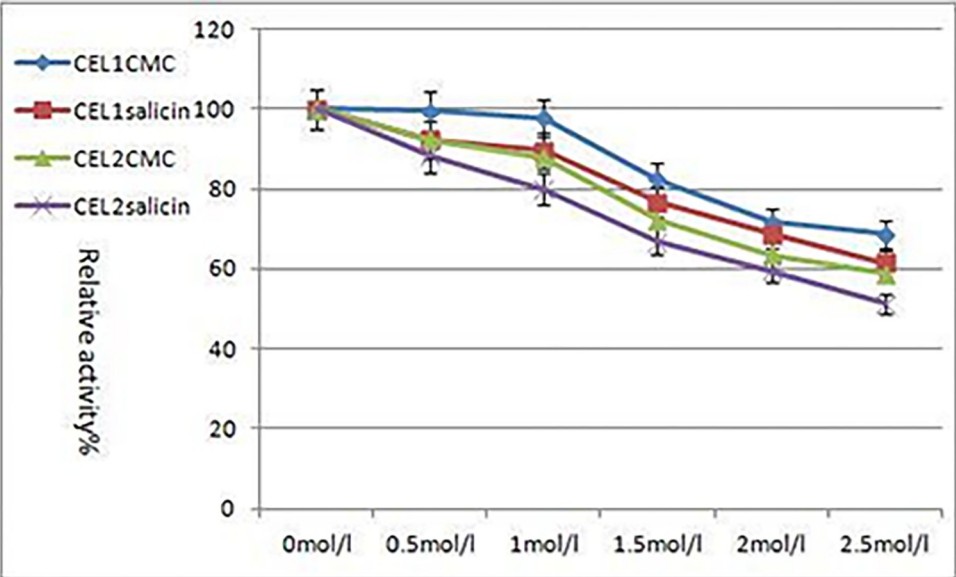

**Fig 10. Salt tolerance of enzyme.**

cellulase activity, indicating that these two amino acids may play a role in the enzymatic activity, while the mutation E247A and E267A substantially lost the glucosidase activity, indicating that these two amino acids may play a crucial role in the enzymatic activity.

As can be seen from Fig 11, the carboxymethylcellulose sodium (CMC-Na) activity of the mutation N68D and N112D was increased; while the E247A, E267A, E351A activity remained almost the same when compared with the wild-type protein; the decrease in carboxymethylcellulose sodium (CMC-Na) activity of Y150H was relatively large. Y150H accounted for 39.67% of CEL1.CEL2 accounted for 216.79% of CEL1. In other words, its activity after truncation is

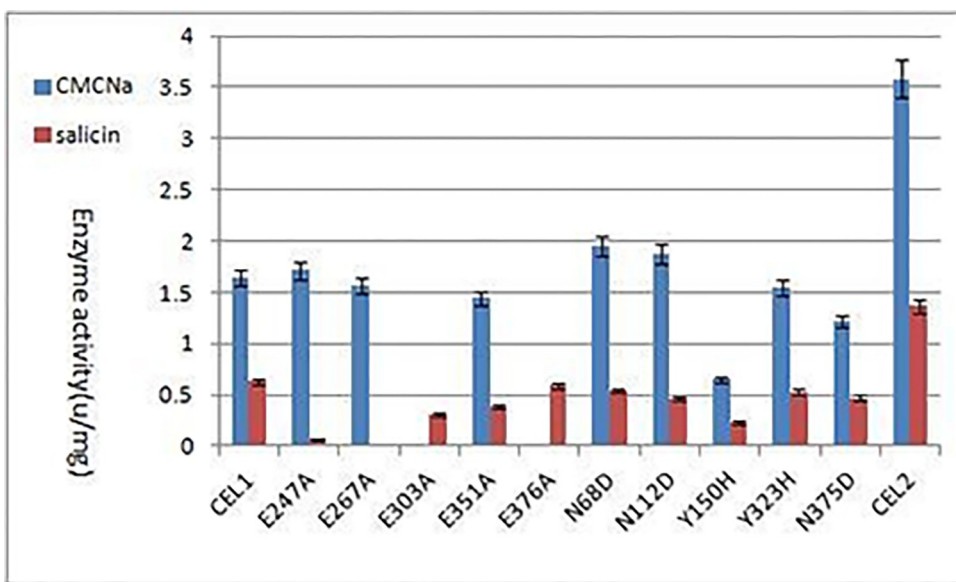

**Fig 11. Comparison of mutant enzyme activities.**

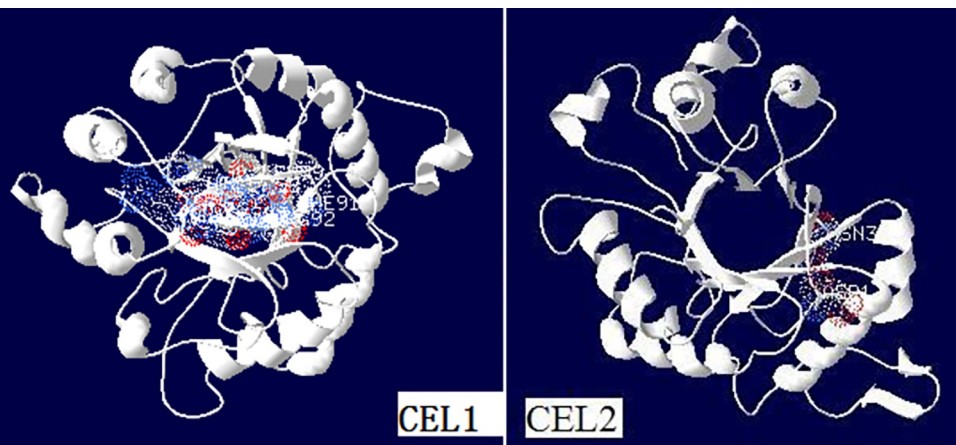

**Fig 12. The 3D structure of CEL1 and CEL2 by computer modeling.**

2.17 times higher than that of CEL1, which is rather significant. The reason for the increase can be analyzed by comparing the 3D structures of the proteins.

The amino acid sequences of CEL1 and CEL2 were submitted to the Phyre2 protein online analysis server [28], and the simulated three-dimensional structure is shown in Fig 12. It was found that there was an obvious pocket structure in the three-dimensional structure of CEL2 protein, which is composed of $(\beta / \alpha)_8$ barrel-shaped folded structures. The opening of the three-dimensional pocket structure of the CEL1 protein was blocked by multiple amino acids, and CEL2 was formed by removing multiple amino acids from the front end of CEL1. This may be because after the removal of multiple blocking amino acid mutations in the import region, the substrate was more likely to enter and bind to the active center of the enzyme, and the enzyme can then easily decompose the substrate. As a result, the activity of CEL2 was further increased by 2.17 times compared to CEL1.

Original amino acids in the mutant protein and the mutated amino acids were analyzed in Missense3D (http://www.sbg.bio.ic.ac.uk/~missense3d/). The changes after the mutation were also analyzed to account for changes in the biological activity of the mutant proteins. It was found that after Glu was mutated to Ala, the uncharged residue Ala typically substituted the charged residue Glu in the protein, which would destroy all side-chain H bonds and/or main-chain H bonds formed by the GLU residues, as shown in Fig 13. For example, after the 376th Glu of CEL1 was substituted by Ala, the H bonds were broken, which further changed the interaction effect between amino acids and affected the biological activity of enzymes. E247A, E267A, and E303A were identical.

As shown in Fig 13, some Glu remained the same after mutation to Ala. After Glu was mutated to Ala in E351A, no H bonds were broken. The reason may be that Glu was located outside the proteins, and did not form H bonds with amino acids outside. The mutation to Ala did not affect the interaction effects between amino acids, so Missense3D analysis showed no change in structure. At the same time, it was discovered that the activity of E351A had little change, as shown in Fig 13, which can be mutually corroborated.

After other amino acids mutated in the CEL1 protein, their biological activities somewhat changed. The 112th Asn (N) was mutated to Asp (D), showing that the charged residue Asp (D) substituted the uncharged residue Asn (N) in the protein, changing the structure and increasing the activity of N112D, as shown in Fig 11. The 375th Gly (G) was mutated to Asp (D), showing that the charged residues substituted the uncharged residues in the protein,

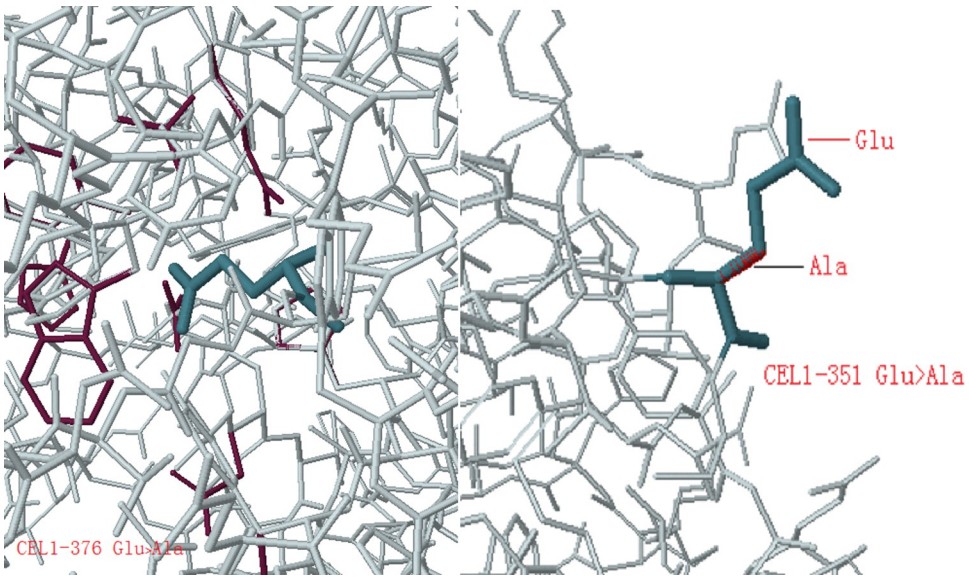

**Fig 13. Missense3D analyzes Glu replaced by Ala.**

which contracted the volume, resulting in a pocket, or a cavity, on the surface. The activity of N375D was lower than the original activity, as shown in Fig 11. Both N112D and N375D were mutated to Asp (D). In Fig 14, N112D showed that Asp (D) had more side chains than Asn (N), which may be more likely to interact with substrates or other amino acids. Mutation of 375th Gly (G) to Asp (D) resulted in shrinkage of cavity volume, which may affect the 376th Glu and reduce its activity.

As shown in Fig 15, the Missense3D analysis showed that the 150th Tyr (Y) in CEL1 protein became His (H), showing that the charged residue His (H) substituted the uncharged residue Tyr (Y) in the protein, which would destroy all side chains/side-chain H bonds and side-chain/main-chain H bonds formed by Tyr (Y) residues. The activity of Y150H was reduced, as can be seen from Fig 11. The 323rd Tyr (Y) changed to His (H), showing no changes in structure. This may be because Tyr (Y) was located outside the protein, and the mutation did not

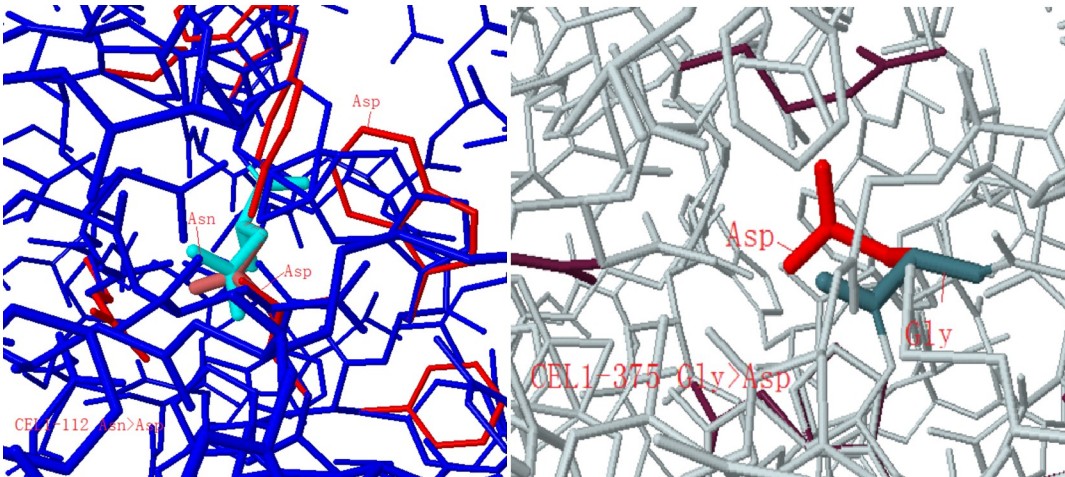

**Fig 14. Missense3D analyzes amino acid replaced by Asp.**

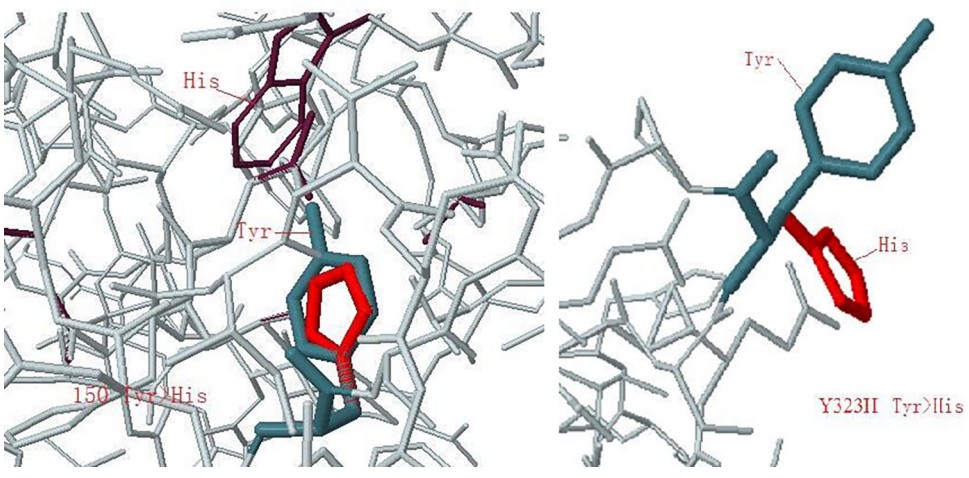

**Fig 15. Missense3D analyzes Tyr replaced by his.**

change the reaction between amino acids. Therefore, the activity of Y323H remained basically the same with little change, as shown in Fig 11.

## Discussion

The gene sequence of *Thermoascus aurantiacus* was retrieved in the GenBank database of NCBI. The size of the target gene *cel1* is about 1365 bp. The gene sequence was obtained after gene sequencing.

The nucleotide sequence of the target *cel1* was analyzed by the blast of NCBI. It was inferred that it possesses endoglucanase properties and falls under the GH5 family [29]. It was also found to have multiple domains that are similar to the nucleotide sequences of glucosidase. It was inferred that *cel1* may possess glucosidase properties, indicating that *cel1* has multifunctionality.

Then, *cel1* was cloned into gene *cel1* to construct an expression vector. The engineered bacteria were obtained after electroporation. After fermentation, the activity of the secreted protein was tested and showed that the enzymatic activity of carboxymethylcellulose sodium (CMC-Na) and salicin were 1.37 U/ml and 0.47 U/ml, respectively. There was little salicin enzymatic activity, less than one-third of that of carboxymethylcellulose sodium (CMC-Na). Through biological information analysis and experimental verification, it can be inferred that *cel1* is endoglucanase of the GH5 family, which is a bifunctional enzyme with glucosidase activity.

In general, fungi with bifunctional activity do not have high enzymatic activity, sharing a catalytic domain. The biological activity of CEL1 is also not high. For example, CbXynl0C in cellulolytic bacteria (*Caldicelluosiruptor bescii*) is a bifunctional xylanase/cellulase enzyme with only one low activity catalytic domain [30].

Through analysis of bioinformatics websites and software, glutamates at specific positions in CEL1 were site-directed mutated to alanine. It was found that Glu247, Glu267, Glu303, and Glu376 may play a role in enzymatic activity. Mutation of the 112th Asn (N) to Asp (D) increased the biological activity of CEL1, and the charged residues substituted the uncharged residues in the protein, which changed the interaction effects between amino acids, thereby affecting the biological activity of the enzymes. Asp (D) played an important role in the hydrolyzation of Asp (D) cellulose, which has also been reported in other families, such as the GH6, GH45, and GH74 families [31, 32].

By comparing the activities of CEL1 and CEL2, it was found that the activity of CEL2 increased by 2.17 times. After removing multiple blocking amino acid mutations in the import region, CEL2 is obtained, as shown in Fig 12. The three-dimensional structure of CLE2 has a very significant hollow structure [33]. It allows better contact with the substrate, thereby increasing the enzymatic activity [34]. Furthermore, the amino acids of CEL1 have a glycosylation site that binds to the polysaccharide chain. The presence of polysaccharide chain repels substrate close to the enzyme, and after the sugar chain is removed, the substrate is more accessible to the enzyme. Therefore, by removing the sugar chain and amino acid sequence, the hollow structure of the protein is exposed, and the substrate can be more easily bound to CEL2. Therefore, the activity of CEL2 is significantly improved compared with that of CEL1. In this way, highly active proteins can be obtained.

## Supporting information

**S1 Fig. Unprocessed version cel1 is the S1 Fig title.** The raw image of Fig 1 in the manuscript is the S1 Fig legend.
(TIF)

**S2 Fig. Unprocessed version cel2 is the S2 Fig title.** The raw image of Fig 1 is the S2 Fig legend.
(TIF)

**S3 Fig. Unprocessed version 2 is the S3 Fig title.** The raw image of Fig 3C in the manuscript is the S3 Fig legend.
(TIF)

**S4 Fig. Unprocessed version 1 is the S4 Fig title.** The raw image of Fig 3C in the manuscript is the S4 Fig legend.
(TIF)

**S5 Fig. Unprocessed version is the S5 Fig title.** The raw image of Fig 4 in the manuscript is the S5 Fig legend.
(TIF)

**S6 Fig. Unprocessed version 1 is the S6 Fig title.** The raw image of Fig 5 in the manuscript is the S6 Fig legend.
(TIF)

**S7 Fig. Unprocessed version 2 is the S7 Fig title.** The raw image of Fig 5 in the manuscript is the S7 Fig legend.
(TIF)

**S1 File. Supplementary material unprocessed gels image is the S1 File title.**
(PDF)

## Author Contributions

**Writing – original draft:** Hongwei Shao.

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
