## [Decision Letter · Decision Letter 0]

3 Nov 2022

PONE-D-22-24693Cloning, Expression, and Molecular Modification of Glycoside Hydrolase Family 5 Genes from Thermoascus aurantiacusPLOS ONE

Dear Dr. Shao,

Thank you for submitting your manuscript to PLOS ONE. After careful consideration, we feel that it has merit but does not fully meet PLOS ONE’s publication criteria as it currently stands. Therefore, we invite you to submit a revised version of the manuscript that addresses the points raised during the review process.

We look forward to receiving your revised manuscript.

Kind regards,

Heping Cao, PhD

Academic Editor

PLOS ONE

Journal Requirements:

Additional Editor Comments:

Please revised your manuscript according to the reviewers' comments and suggestions.

Reviewers' comments:

Reviewer's Responses to Questions

**Comments to the Author**

1. Is the manuscript technically sound, and do the data support the conclusions?

Reviewer #1: Partly

Reviewer #2: Yes

2. Has the statistical analysis been performed appropriately and rigorously? 

Reviewer #1: N/A

Reviewer #2: I Don't Know

3. Have the authors made all data underlying the findings in their manuscript fully available?

Reviewer #1: Yes

Reviewer #2: Yes

4. Is the manuscript presented in an intelligible fashion and written in standard English?

Reviewer #1: Yes

Reviewer #2: No

5. Review Comments to the Author

Reviewer #1: The manuscript, entitled “Cloning, Expression, and Molecular Modification of Glycoside Hydrolase Family 5 Genes from Thermoascus aurantiacus”, reported the overexpression, purification, and characterization of an endoglucanase enzyme from Thermoascus aurantiacus IMI. Bioinformatic data from the manuscript showed that the fungal enzyme is highly homologous to other endoglucanase enzymes. Using purified recombinant proteins, the author showed that the Cel1 protein has GH5 endoglucanase activity. While the work of endoglucanase from other strains of Thermoascus aurantiacus has already been reported years ago, constructions of the cel1 mutants gene and the study of the truncated version of the cel1 gene, called cel2, were not studied before. Data showed that the molecularly engineered Cel2 protein has an enhanced activity of GH5 endoglucanase. However, the manuscript has several shortcomings that need to be addressed before its publication on PLOS One.

Major comments:

Line 89: the gene sequence of the cel1 gene (JC988729.1) was downloaded from GenBank. However, it is not clear if the gene sequence belongs to T. aurantiacus IMI strain.

Line 151 and Figure 3: The recombinant proteins were purified by either DEAE-Sepharose column or Purification by Ni-NTA affinity chromatography. Since the DEAE-Sepharose is a weak anion exchange chromatography resin, it is interesting to see that there was only one protein band on SDS-PAGE. Also, the purified protein band didn’t have degradation products. I recommend that the author revises figure 3 to include lanes of overexpression and purification procedure. Specifically, uninduced and induced P. pastoris GS115 and chromatography wash fraction.

Figures 3 and 4: the nature of overexpressed Cel1 and Cel2 proteins is unknown. The purified proteins in Figure 3 and Figure 4 may be Cel1 and Cel2 proteins but could be something else. Additional assays, such as Western blot or mass spec, are needed to confirm that the purified proteins are Cel1 and Cel2 proteins.

Lines 159-160, Line 165, Line 194, and Line 203: The concentration of the recombinant protein was not stated in the assays since only 100 uL of the enzyme solution was stated.

Minor comments:

Lines 82-82: The statement “the cDNA of T. aurantiacus was obtained by reverse

transcription polymerase chain reaction (RT-PCR )” is a little confusing. Please revise the purpose of doing RT-PCR.

Lines 143-145: It was not clear what the inducer is for overexpression of the recombinant proteins.

Lines 185-191: Did the assay have sample replicates in determining the Michaelis constant Km of the enzymes? Also, for Lines 212-218, Lines 220-226, and Lines 228-234: The same comment if the assays have sample replicates for determining the effects of different metal ions on enzyme activity, effects of different organic solvents on enzyme activity, and effects of salt tolerance.

Lines 321-324: The manuscript reported that the CEL1 enzyme is not resistant to high temperatures above 65�C. However, another study reported that the enzyme from another strain of Thermoascus aurantiacus has an optimum temperature of 70�C.

Line 339 and Figure 7: The author reported that the CEL1 enzyme’s activity was dramatically decreased when the pH is greater than pH5 and when the pH is lower than pH5. However, another study reported that the enzyme from another strain of Thermoascus aurantiacus is stable over a broad pH range of 3.0-7.0.

Line 238: SPSS method was stated; however, the manuscript didn’t report any statistical results.

The overall style of the manuscript: starting on Page 25, the text of the manuscript changed the style from left-aligned to center-aligned.

Reviewer #2: The manuscript is too lengthy and not written in a fashion others can understand. The authors should get professional help in writing so that it could be understood and interpreted. I suggest that it should be re-written and submitted.

6. PLOS authors have the option to publish the peer review history of their article (what does this mean?). If published, this will include your full peer review and any attached files.

Reviewer #1: **Yes: **Jianmin Zhong

Reviewer #2: No

---

## [Author Response · Author response to Decision Letter 0]

26 Feb 2023

See the document for modification.For example，The gene sequence of the cel1 gene (JC988729.1) was downloaded from GenBank. The following information is available.SOURCE ：Thermoascus aurantiacus ORGANISM：Thermoascus aurantiacus Eukaryota; Fungi; Dikarya; Ascomycota; Pezizomycotina;Eurotiomycetes; Eurotiomycetidae; Eurotiales; Thermoascaceae;Thermoascus.

TITLE：POLYPEPTIDES HAVING ENDOGLUCANASE ACTIVITY AND POLYNUCLEOTIDES ENCODING SAME

JOURNAL：Patent: EP 2791329-A1 17 22-OCT-2014;NOVOZYMES INC [US]

---

## [Decision Letter · Decision Letter 1]

28 Apr 2023

Cloning, Expression, and Molecular Modification of Glycoside Hydrolase Family 5 Genes from Thermoascus aurantiacus

PONE-D-22-24693R1

Dear Dr. Shao,

We’re pleased to inform you that your manuscript has been judged scientifically suitable for publication and will be formally accepted for publication once it meets all outstanding technical requirements.

Kind regards,

Heping Cao, PhD

Academic Editor

PLOS ONE

Additional Editor Comments (optional):

Reviewers' comments:

Reviewer's Responses to Questions

**Comments to the Author**

1. If the authors have adequately addressed your comments raised in a previous round of review and you feel that this manuscript is now acceptable for publication, you may indicate that here to bypass the “Comments to the Author” section, enter your conflict of interest statement in the “Confidential to Editor” section, and submit your "Accept" recommendation.

Reviewer #2: All comments have been addressed

2. Is the manuscript technically sound, and do the data support the conclusions?

Reviewer #2: Yes

3. Has the statistical analysis been performed appropriately and rigorously? 

Reviewer #2: Yes

4. Have the authors made all data underlying the findings in their manuscript fully available?

Reviewer #2: Yes

5. Is the manuscript presented in an intelligible fashion and written in standard English?

Reviewer #2: Yes

6. Review Comments to the Author

Reviewer #2: The authors cloned a bifunctional cellulase gene cel1 from Thermoascus aurantiacu s in Pichia pastoris. They studied in detail the structure, function and relational aspects of the expressed enzyme. The bioinformatics data strengthens their experimental data. Hence, I recommend publication of this manuscript in PLOS.

7. PLOS authors have the option to publish the peer review history of their article (what does this mean?). If published, this will include your full peer review and any attached files.

Reviewer #2: No

---

## [Editor Report · Acceptance letter]

8 May 2023

PONE-D-22-24693R1 

Cloning, Expression, and Molecular Modification of Glycoside Hydrolase Family 5 Genes from *Thermoascus aurantiacus*

Dear Dr. Shao:

I'm pleased to inform you that your manuscript has been deemed suitable for publication in PLOS ONE. Congratulations! Your manuscript is now with our production department. 

Kind regards, 

on behalf of

Dr. Heping Cao 

Academic Editor

PLOS ONE